# Loss of SP-A in the Lung Exacerbates Pulmonary Fibrosis

**DOI:** 10.3390/ijms23105292

**Published:** 2022-05-10

**Authors:** Kyunghwa Kim, Dasom Shin, Gaheon Lee, Hyunsu Bae

**Affiliations:** 1Department of Health Sciences, The Graduate School of Dong-A University, 840 Hadan-dong, Saha-gu, Busan 49315, Korea; kyungkim@dau.ac.kr (K.K.); jingun1984@naver.com (G.L.); 2Department of Physiology, College of Korean Medicine, Kyung Hee University, 26-6 Kyungheedae-ro, Dongdaemoon-gu, Seoul 02453, Korea; ssd060@naver.com

**Keywords:** surfactant protein A, pulmonary fibrosis, transforming growth factor-β1, CRISPR/Cas9

## Abstract

Idiopathic pulmonary fibrosis (IPF) is a devastating and common chronic lung disease that is pathologically characterized by the destruction of lung architecture and the accumulation of extracellular matrix in the lung. Previous studies have shown an association between lung surfactant protein (SP) and the pathogenesis of IPF, as demonstrated by mutations and the altered expression of SP in patients with IPF. However, the role of SP in the development of lung fibrosis is poorly understood. In this study, the role of surfactant protein A (SP-A) was explored in experimental lung fibrosis induced with a low or high dose of bleomycin (BLM) and CRISPR/Cas9-mediated genetic deletion of SP-A. Our results showed that lung SP-A deficiency in mice promoted the development of fibrotic damage and exacerbated inflammatory responses to the BLM challenge. In vitro experiments with murine lung epithelial LA-4 cells demonstrated that in response to transforming growth factor-β1 (TGF-β1), LA-4 cells had a decreased protein expression of SP-A. Furthermore, exogenous SP administration to LA-4 cells inhibited the TGF-β1-induced upregulation of fibrotic markers. Overall, these findings suggest a novel antifibrotic mechanism of SP-A in the development of lung fibrosis, which indicates the therapeutic potential of the lung SP-A in preventing the development of IPF.

## 1. Introduction

Idiopathic pulmonary fibrosis (IPF) is a chronic, progressive, and fatal interstitial lung disease [1]. IPF is characterized by exertional dyspnea and pulmonary dysfunction, and has a median survival of only 3–5 years after diagnosis [2]. Generally, lung damage due to pulmonary fibrosis is irreversible with significant morbidity and mortality. Thus, diagnosis and intervention at the earliest possible stage are successful therapeutic strategies against IPF. However, the detailed mechanisms of the development of lung fibrosis are still unclear.

Several potential diagnostic and prognostic biomarkers of IPF have been reported in patients with IPF [3]. Among them, surfactant protein A (SP-A) levels are clearly changed in the serum and bronchoalveolar lavage (BAL) fluid of IPF patients [4]. Furthermore, an early increase in SP-A was highly correlated with an increased risk of mortality in IPF patients [5]. Interestingly, the difference in the protein level of SP-A in IPF patients was tissue specific. In contrast to the high SP-A level in the blood, SP-A in BAL fluid was decreased in IPF patients, and the level clearly correlated with survival in patients with IPF [6]. Beyond alterations in gene expression, mutations in SP-A are also strongly associated with IPF [7]. These data suggest the potential of SP-A as a predictor of mortality and the progression of IPF.

SP-A plays a crucial role in maintaining lung homeostasis and the innate immune response [8]. SP-A, which is a hydrophilic protein, is synthesized by airway cells, primarily secreted by alveolar epithelial type II pneumocytes, and localized within blood vessels [9]. In particular, SP-A is thought to control pulmonary inflammation in diverse ways, including directly binding to pathogens and enhancing phagocytosis. To date, the mechanisms and pathophysiological importance of SP-A in the lung during IPF are unknown. Thus, understanding this association may be important for determining its potential in the clinic as a novel therapeutic target.

Based on these observations, we hypothesized that SP-A in the lung is a key molecule in the development of pulmonary fibrosis. To test this hypothesis, we generated SP-A-deficient mice through clustered regularly interspaced short palindromic repeats (CRISPR)/CRISPR associated protein 9 (Cas9)-mediated gene knockdown. To evaluate the role of SP-A in the initiation and development of lung fibrosis, different levels of lung fibrosis were induced with low or high doses of bleomycin (BLM) on mice. Furthermore, the underlying mechanisms were evaluated in vitro in a murine lung epithelial cell line (LA-4) by exogenous treatment with lung surfactant protein following transforming growth factor beta-1 (TGF-β1) challenge.

## 2. Results

### 2.1. SP-A Levels in the Lung Are Dose-Dependently Altered in Mice Stimulated with Bleomycin

To investigate whether lung SP-A is associated with the prognosis of IPF, we induced mice with a low dose (0.1 mg/kg) or a high dose of BLM (5 mg/kg). As shown in Appendix A, we found lung fibrosis on mice with bleomycin treatment in a dose-dependent manner.

On these BLM-treated mice, we measured the protein levels of SP-A in lung tissue (Figure 1A,B) and BAL fluid (Figure 1C) and further mRNA levels of SP-A in lung tissue (Figure 1D). Importantly, high-dose BLM-induced mice exhibited clear reductions in SP-A levels in both lung and BAL fluid compared to PBS-injected control mice. Therefore, we investigated the association of SP-A levels and BLM-induced lung fibrosis by measuring the protein levels of fibrotic markers, such as α-smooth muscle actin (α-SMA). As expected, 5 mg/kg BLM-injected mice exhibited increased α-SMA. These data suggest a negative correlation between α-SMA and SP-A on BLM-induced mice. Furthermore, we found a loss of epithelial markers, such as E-cadherin, on high-dose BLM-induced mice, suggesting a positive correlation between E-cadherin and SP-A. Notably, a low dose of BLM induced no differences in lung SP-A levels compared to those in the control group.

### 2.2. SP-A-Deficient Mice Displayed Enhanced BLM-Induced Lung Fibrosis

We sought to explore the role of SP-A in the pathogenesis of IPF in SP-A-deficient mice using the CRISPR/Cas9 system (Figure 2A). As shown in Figure 2B, we observed a substantial loss of SP-A protein expression in the lung tissues of SP-A-deficient mice compared to wild-type (WT) mice injected with a lentivirus expressing nontargeting sgRNA. A real-time qPCR analysis further confirmed the specific knockout of SP-A at the mRNA level in SP-A-deficient mice (Figure 2C). SP-A knockout using CRISPR/Cas9-mediated genetic editing caused no significant alterations in the expression of other surfactant proteins.

After confirming CRISPR knockout, we next investigated the role of lung SP-A in the development of pulmonary fibrosis with BLM application. As expected, wild-type mice injected with a low dose of BLM (0.1 mg/kg) displayed no detectable collagen deposition and architectural destruction in the lungs (Figure 2D,E). However, the level of lung fibrosis was dramatically enhanced in SP-A-deficient (SP-A^KO^) mice treated with a low dose of BLM. On mice induced with a high dose of BLM, no significant difference was detected between SP-A^KO^ mice and WT mice. Importantly, these enhanced fibrotic features in SP-A^KO^ mice were evaluated by Western blot analysis (Figure 2F,G). A high dose of BLM, not a low dose of BLM, seemed to induce increased expression of α-SMA and decreased expression of E-cadherin. These altered expression levels were not significantly different between SP-A^KO^ mice and WT mice. However, low doses of BLM-treated SP-A^KO^ mice clearly showed upregulation of α-SMA and downregulation of E-cadherin compared to those of WT mice.

### 2.3. SP-A Deficiency Increases Inflammatory Accumulation in Mice Induced with a Low Dose of Bleomycin

During the pathogenesis of IPF, pulmonary inflammation is thought to develop and orchestrate fibrotic responses. Here, we explored whether SP-A deficiency had an effect on pulmonary inflammation in BLM-challenged mice. The influx of inflammatory cells into the lungs was analyzed to assess lung inflammation by performing a counting analysis of BAL fluid. We observed significant accumulation of total inflammatory cells in SP-A^KO^ mice induced with a low dose of BLM compared to control mice (Figure 3A). Inflammatory cells, including macrophages (Figure 3B), neutrophils (Figure 3C), and eosinophils (Figure 3D), were also extensively detected in the lung tissues of SP-A-deficient mice induced with BLM (low dose: 0.1 mg/kg). Under severe lung fibrosis conditions induced with a high dose (5 mg/kg) of BLM, no significant difference was observed following lentivirus delivery. These inflammatory features in SP-A-deficient mice were further assessed by the histological evaluation (Figure 3E) and semi-quantitative histological analysis (Figure 3F) of H&E-stained lung sections.

### 2.4. SP-A Deficiency Causes Excessive Expression of Fibrotic Genes and Inflammatory Cytokines in Low-Dose BLM-Induced Mice

Next, we explored whether SP-A deficiency affected the expression of pivotal genes associated with fibrogenesis, including TGF-β1, collagen, and fibronectin (Fn1). Importantly, we observed excessive mRNA levels of TGF-β1 in the lung tissues of SP-A-deficient BLM-induced mice (low dose of BLM: 0.1 mg/kg) (Figure 4A). Similarly, we observed the upregulation of collagen III (Col3A1) (Figure 4B) and Fn1 (Figure 4C) in low-dose BLM mice following SP-A knockout. However, we found no significant difference among the high-dose BLM-induced groups (SP-A^KO^ mice vs. wild-type mice).

Next, we examined whether an SP-A deficiency caused alterations in the protein levels of proinflammatory cytokines in the lung tissues of mice. Indeed, we observed a notable increase in proinflammatory cytokines, including TNF-α, IFN-γ, and IL-6, but not IL-23 in SP-A-deficient mice challenged with a low dose of BLM (Figure 4D–G). However, an SP-A deficiency produced no clear alterations in the levels of proinflammatory cytokines in high-dose BLM-induced mice.

### 2.5. TGF-β1 Inhibits SP-A in Lung Epithelial Cells

TGF-β1 signaling has been postulated to be a hub pathway during the development and progression of pulmonary fibrosis. Although the detailed mechanisms are still unclear, TGF-β1 activation induces the excessive deposition of extracellular matrix by controlling the expression of genes associated with fibrogenesis. In this study, we examined whether TGF-β1 affected the expression of SP-A in vitro in a murine lung epithelial cell line (LA-4) that shows the characteristics of type II pneumocytes.

Indeed, TGF-β1 treatment induced LA-4 cells to undergo epithelial–mesenchymal transition (EMT), a process whereby lung epithelial cells obtain a mesenchymal phenotype during fibrotic processes (Figure 5A). TGF-β1 clearly induced dose-dependent morphological alterations from oval epithelial cells to spindle-shaped fibroblast-like cells. Importantly, we noted a significant dose-dependent reduction in SP-A expression in LA-4 cells in response to TGF-β1 treatment (Figure 5B,C). EMT induced by TGF-β1 treatment was confirmed with a Western blot of the upregulation of E-cadherin and downregulation of α-SMA.

### 2.6. SP-A Blocks TGF-β1-Induced EMT in Alveolar Epithelial LA-4 Cells

To address the functional importance of SP-A in lung fibrosis, we administered lung surfactant protein and TGF-β1 to LA-4 cells. As shown in Figure 6A, co-administration of TGF-β1 (20 μg/mL) and SP inhibited TGF-β1-induced morphological alteration in LA-4 cells in a dose-dependent manner. Interestingly, exogenous treatment of SP on LA-4 cells effectively reversed downregulation of SP-A following TGF-β1 challenge (Figure 6B,C). As expected, 20 μg/mL TGF-β1 induced a reduction of E-cadherin and an increase in α-SMA. However, co-administration of SP and TGF-β1 suppressed these TGF-β1-dependent changes of gene expressions.

## 3. Discussion

In this study, we demonstrated the beneficial effects of SP-A against lung fibrosis in vivo and in vitro. CRISPR/Cas9-mediated deletion of SP-A worsened BLM-induced lung fibrosis and lung dysfunction. The expression of SP-A in the lung inversely correlated with disease severity in pulmonary fibrosis, as evidenced by the effects of different doses of BLM on mice. Mechanistic investigations demonstrated that SP-A protects against lung fibrosis by regulating inflammatory responses in the lung during fibrogenesis. In lung epithelial cells, SP-A protein levels were decreased in response to TGF-β1 stimulation. Importantly, lung surfactant protein treatment reversed TGF-β1-induced EMT. Taken together, these data suggest the therapeutic potential of SP-A in the lung in patients to target the development of pulmonary fibrosis. Many studies have investigated the identity of a predictive biomarker for disease progression against idiopathic pulmonary fibrosis [10].

Notably, alveolar epithelial markers have been reported as potential molecular biomarkers that reflect biological effects in response to lung injury. Among them, we focused on SP-A, which is an abundant alveolar protein synthesized and extensively processed by type II cells during fibrogenesis. Indeed, alveolar epithelial type II (AT2) cells have essential roles in the alveolus to maintain lung homeostasis [11]. Several intrinsic and environmental factors are linked to the functional outcomes of AT2 cells. It is not surprising that a large number of factors that accumulate with age induce AT2 dysfunction by perturbing lung homeostasis [12]. Impairment of these cells results in a reduction in surface tension force and fibrotic fusion of alveolar basement membranes, which contributes to the development of IPF. In particular, mounting evidence suggests AT2 cell dysfunction as an early process in IPF that initiates the development of fibrosis, including the production of mediators involved in the proliferation and activation of fibroblasts and excessive secretion of extracellular matrix proteins [13]. Currently, the mechanistic details of the vicious cycle of multidirectional interactions between the epithelium and other cells remain unclear.

SP-A concentrations are altered in patients with IPF. Cormack et al. [14] and Gunther et al. [15] reported significantly lower levels of SP-A in the BAL fluid of IPF patients than in healthy subjects. However, circulating SP-A in the blood was elevated in IPF patients, and levels of the serum SP-A correlated with the progression of fibrosis in IPF patients [16]. Another surfactant protein, surfactant protein D (SP-D), has also been reported to be a predictive biomarker of IPF [17]. Interestingly, despite similarities in structure, SP-A and SP-D have different binding specificities and secretion characteristics. In IPF patients, leakage of SP-D from the alveolar space into the blood was higher than SP-A in response to lung damage [17]. However, the detailed mechanisms of the production and secretion of SP-A and SP-D in IPF are still unclear. In this study, we measured the protein levels of SP-A in lung tissue and BAL fluid in a fibrosis mouse model induced with low and high doses of BLM. A low dose of BLM (0.1 mg/kg) elicited no robust toxicity in mice, as assessed by survival rate, histological images, and body weight. By contrast, mice induced with a high dose of BLM (5 mg/kg) displayed severe lung damage and fibrotic alterations in the lung with toxicity. Importantly, we found a significant loss of SP-A in both the lung and BAL fluid of mice with IPF-associated toxicity (5 mg/kg BLM-induced mice). However, the level of SP-A was not significantly different in mice induced with a low dose of BLM compared to control mice. Furthermore, CRISPR/Cas9-mediated SP-A knockout resulted in the development of fibrotic damage and enhanced inflammatory responses specifically in low-dose BLM-induced mice. Based on these observations, we believe that SP-A is functionally involved in the lung, mainly during the early inflammatory stages before the fibrotic stages occur. Furthermore, we believe that the level of SP-A in lavage and lung tissue can be representative of the functionality and regenerative capacity of lung epithelial cells, which maintain lung homeostasis in response to harmful stimuli and protect against lung fibrosis. Further studies are needed to clearly elucidate these relationships between lung SP-A levels and IPF disease progression.

TGF-β1 is a multifunctional cytokine that regulates the function of alveolar cells and mediates fibrotic responses [18]. Many studies have evaluated the potential associations between mutant SP-A and familial pulmonary fibrosis [19]. The underlying mechanisms remain unclear, but TGF-β1 signaling is postulated to be an essential factor linking this association between SP-A and IPF. In this study, consistent with prior results [20], we observed a reduction in SP-A in lung epithelial alveolar cells following TGF-β1 administration. Additionally, lung surfactant treatment not only reversed TGF-β1-induced EMT changes but also inhibited the expression of genes associated with fibrogenesis in lung epithelial LA-4 cells. It is likely that in response to pulmonary insult, TGF-β1 induces EMT changes in lung epithelial cells, which results in a decrease in SP-A protein expression. The excessive secretion of TGF-β1 may initiate a vicious cycle that alters the expression of genes and enhances inflammatory responses in the pathogenesis of IPF. Based on this evidence, we believe that exogenous treatment with SP-A contributes to maintaining lung function, subsequently inhibiting fibrosis to protect against the effects of TGF-β1.

There were several limitations in the current study. Firstly, we investigated the impact of SP-A on BLM-induced lung fibrosis without further evaluation of the underlying mechanisms. Secondly, we performed biochemical experiments only 14 days post BLM. Thus, the impact of SP-A on lung fibrosis could be different based on the developing phases of lung fibrosis. Thirdly, we treated SP protein which was purified from mouse BAL fluid on lung epithelial cells against TGF-β1. The therapeutic effect of synthetic SP-A could be different from that of isolated SP protein.

In conclusion, this study demonstrated that SP-A loss exacerbated lung fibrosis induced by BLM in mice. The loss of SP-A protein expression was characterized by fibrotic change in the lung on BLM-treated mice. In addition, we also demonstrated that exogenous SP treatment effectively reversed TGF-β1-induced EMT in vitro. These data have significant implications for future efforts in developing a novel therapeutic strategy for treating IPF by targeting SP-A functions. Future experiments are needed to understand the mechanisms underlying the impact of SP-A against IPF.

## 4. Materials and Methods

### 4.1. Animals

All experiments were performed in accordance with animal protocols and guidelines approved by the Animal Experimental Ethics Committee of Kyung Hee University and Dong-A University. For the CRISPR/Cas9-mediated genetic model, Rosa26-Cas9 knock-in mice were obtained from the Jackson Laboratory (stock no: 024858). C57BL6/J mice purchased from Daehan Biolink (DBL, Seoul, Korea) were used to generate a mouse model of IPF using bleomycin induction.

### 4.2. Cells

Human embryonic kidney (HEK) 293 cells and murine lung alveolar epithelial LA-4 cells were obtained from the Korean Cell Line Bank for Biological Sciences (KCLB, Seoul, South Korea). HEK293 cells were cultured in Dulbecco’s modified Eagle’s medium (DMEM) (ThermoFisher Scientific, Waltham, MA, USA) supplemented with 10% fetal bovine serum (FBS) and 1% penicillin–streptomycin (PS) (ThermoFisher Scientific, Waltham, MA, USA). LA-4 cells were grown in Ham’s F-12K medium (ThermoFisher Scientific, Waltham, MA, USA) with 15% FBS and 1% PS.

### 4.3. Lentiviral Construct Generation

The lentiGuide-Puro plasmid containing a puromycin-resistance cassette (Addgene plasmid # 52963) was used as the backbone for constructing the lenti-sgRNA against the SP-A plasmid as previously described [21]. The guide sequence was selected with the CRISPR/Cas9 target online predictor (CCTop, http://crispr.cos.uni-heidelberg.de/index.html, accessed on 1 May 2019). Pairs of oligos with sgRNA sequences (5′-AGCAATGTGGCCACCGGCTC-3′, 5′-GAGCCGGTGGCCACA-TTGCT-3′), including BsmBI restriction site overhangs targeting SP-A, were annealed and cooled down. The annealed oligonucleotides were purchased, 5′-phosphorylated, and ligated into lentiGuide-Puro using T4 ligase (Thermo Scientific Scientific, Waltham, MA, USA) according to the manufacturer’s instructions. The ligation mix was transformed using competent cells (New England BioLabs, Ipswitch, MA, USA). Correct insertion of the sgRNA sequence was evaluated by Sanger sequencing.

Lentivirus packaging was carried out as previously described with minor modifications [22]. Briefly, lenti-sgRNA plasmids were cotransfected with psPAX2 and pCMV-VSV-G into HEK293 cells using Lipofectamine 2000 (ThermoFisher Scientific, Waltham, MA, USA). pCMV-VSV-G was a gift from Bob Weinberg (Addgene plasmid # 8454). psPAX2 was a gift from Didier Trono (Addgene plasmid # 12260). At 48 h post transfection, the viral-containing supernatant was collected through a 0.45 µm filter (Sigma, St Louis, MO, USA) and concentrated according to the manufacturer’s instructions. These vectors were then titered as previously described [23]. Concentrated lentiviruses were stored at −80 °C until use.

### 4.4. In Vivo Transduction

To transduce the lentivirus, mice were anesthetized with isoflurane. Concentrated lentivirus (100 μL) was delivered via an 18-gauge needle into the posterior oropharynx above the tracheal entrance as previously reported [24]. To increase transduction efficiency, lentiviral supernatants were mixed with Lipofectamine 2000 (ThermoFisher Scientific, Waltham, MA, USA) before instillation at a final concentration of 5%. Lentivirus was delivered before BLM was administered to mice on the same day.

### 4.5. Bleomycin-Induced Pulmonary Fibrosis Mouse Model

To induce pulmonary fibrosis, bleomycin was purchased from Sigma (St Louis, MO, USA) and administered into mice at a final dose of 5 mg/kg or 0.1 mg/kg to generate lung fibrosis [25]. Briefly, the mice were lightly anesthetized, and BLM was dissolved in 40 µL of PBS and intratracheally administered as previously described [26].

### 4.6. Histopathological Staining

According to our previous study [25], lungs were removed from the mice and fixed with 4% paraformaldehyde before being embedded. Serial paraffin sections were prepared using a microtome and then deparaffinized. Paraffin-embedded sections were stained with hematoxylin and eosin (H&E) or Masson’s trichrome (Sigma, St. Louis, MO, USA).

To measure the severity of pulmonary fibrosis, each field was semiquantitatively assessed as previously described with minor modifications [27]. Grading was scored on a scale from 0 to 8 as follows: Grade 0, normal healthy lung; Grade 1, minimal fibrous thickening of bronchiolar or alveolar walls; Grade 2–3, moderate level of thickening of the walls without clear damage to lung architecture; Grade 4–5, severe fibrosis with excessive damage to lung structure and the formation of fibrous bands or fibrous masses; Grade 6–7, severe distortion of lung structure; Grade 8, general fibrous obliteration.

To evaluate the pulmonary inflammation scores of H&E-stained tissues, inflammation was assessed as described previously [28]. Briefly, the score was based on a 13-point scale that measures airway inflammation (4 points), vascular inflammation (4 points), and parenchymal inflammation (5 points). All scoring was performed in a blinded manner.

### 4.7. Western Blotting

Samples of lung tissues and LA-4 cells were homogenized in RIPA buffer (Cell Signaling, Boston, MA, USA) supplemented with a protease inhibitor cocktail (Roche, Basel, Switzerland). Protein samples in BAL fluid were extracted with the trichloroacetic acid (TCA) precipitation method [29]. Then, 30 μg of total protein was separated by gel electrophoresis and transferred to a nitrocellulose membrane (EMD Millipore, Billerica, MA, USA). The membranes were blocked with 5% skim milk in TBST (20 mM Tris-HCl at pH 7.4, 150 mM NaCl, 0.05% Tween-20) for 1 h, followed by incubation with the appropriate primary antibodies overnight at 4 °C. Antibodies against SP-A (#BS-10265R, 1:200) and GAPDH (1:500) were obtained from ThermoFisher Scientific (Waltham, MA, USA). Antibodies against E-cadherin (1:500) and α-SMA (1:500) purchased from Abcam (Cambridge, MA, USA). After the membranes were washed with TBST, they were incubated with horseradish peroxidase-conjugated IgG. HRP-conjugated anti-rabbit and anti-mouse IgG were obtained from Cell Signaling Technology, and anti-goat IgG-HRP antibodies were obtained from Santa Cruz Biotechnology (Dallas, TX, USA). The signals were visualized with enhanced chemiluminescence reagents (Clarity Western ECL Substrate, Bio-Rad, Hercules, CA, USA). The results were analyzed using the ImageJ densitometry system.

### 4.8. Immune Cell Analysis in Bronchoalveolar Lavage Fluid

To collect BAL fluid, 1 mL of PBS was infused into the lungs and extracted via tracheal cannulation two times (harvested BAL fluid volume 1.0–1.5 mL). The collected BAL fluid was centrifuged at 300× *g* for 10 min at 4 °C, the supernatant was removed, and the cell pellet was resuspended in 1 mL of PBS. Next, the total live cell count was determined using a hemocytometer, and BAL cells were attached to glass sides using Cytospin (Sandon, Waltham, MA, USA) and stained with a Diff-Quick staining kit (ThermoFisher, Waltham, MA, USA). BAL fluid cell counts were performed using randomly selected samples in a blinded manner. After Diff-Quick staining, the cells were identified as macrophages, neutrophils, lymphocytes, and eosinophils under light microscopy.

### 4.9. Quantitative Real-Time Polymerase Chain Reaction (PCR)

Total RNA was extracted from lung tissues and cells using Easy BlueTM (Intron Company, Seongnam, South Korea). First-strand cDNA was synthesized with a cDNA synthesis kit (Bioneer Corporation, Daejeon, South Korea) according to the manufacturer’s instructions. Real-time PCR analysis was performed in a LightCycler 96 (Roche, Basel, Switzerland) with SYBR Green I master mix using a SensiFAST™ SYBR^®^ No-ROX Kit (Bioline, Paris, France). The oligonucleotides used are shown in Table 1.

### 4.10. Enzyme-Linked Immunosorbent Assay (ELISA)

Mouse lung tissue proteins were extracted using RIPA buffer with a protease inhibitor cocktail. The levels of the cytokines IL-6, IFN-γ, and TNF-α were analyzed using an ELISA kit from BD Biosciences (San Jose, CA, USA), and IL-23 was analyzed with a kit from R&D Systems (Minneapolis, MN, USA) using a quantitative sandwich method according to the manufacturer’s protocols. The OD (optical density) of each sample after color development was measured with a microplate reader (SOFT max PRO software, Sunnyvale, CA, USA) at 450 nm.

### 4.11. Surfactant Protein Isolation

Surfactant was purified from healthy control mouse lavage fluid as previously described with minor modifications [30]. Briefly, the lavage fluid was centrifuged at 20,000× *g* for 15 h at 4 °C, and the pellets were resuspended in buffer containing 1.64 M sodium bromide. The samples were homogenized and then centrifuged at 60,000× *g* for 4 h at 4 °C in a SWING 28 rotor (Beckman Coulter, Roissy, France). The pellicle was resuspended in a buffer containing 5 mM Tris-HCl (pH 7.4) and 100 mM NaCl (pH 7.4) and then centrifuged twice at 100,000× *g* for 1 h at 4 °C. The resultant pellet was resuspended in PBS and used as surfactant for the treatment of lung alveolar cells.

### 4.12. Statistical Analysis

All statistical analyses of the data were performed using Prism 5 (GraphPad Software Inc., San Diego, CA, USA). All values are shown as the mean ± standard error (SE). The statistical significance of differences between experimental groups was evaluated by the Mann–Whitney U test or unpaired *t* test. *p* values < 0.05 were considered statistically significant.

## Figures and Tables

**Figure 1 ijms-23-05292-f001:**
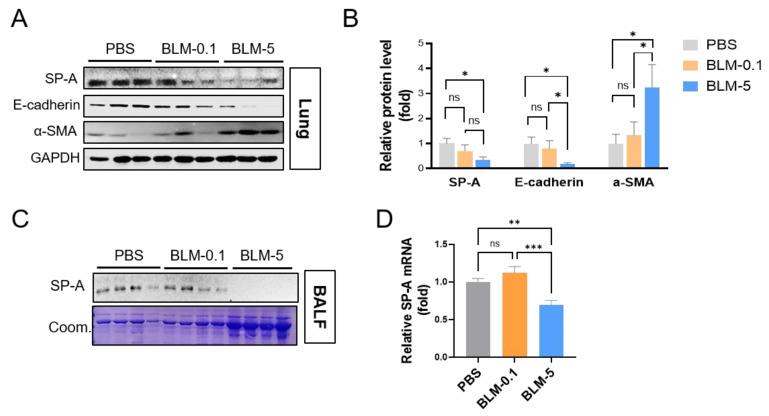
SP-A expression inversely correlates with the severity of pulmonary fibrosis in mice induced with bleomycin. (**A**–**C**) Western blot images showing the expression of SP-A in lung tissues (**A**,**B**) and BAL fluid (**C**) from mice induced with BLM. The relative protein level of E-cadherin or α-SMA normalized to GAPDH was calculated as fold change compared to those in PBS-treated group. Coom. indicates Coomassie staining as loading control. (**D**) Quantitative real-time qPCR analysis of SP-A was used to measure the mRNA levels in the lung tissues in each group. *n* = 5–9 per group. The data are presented as the mean ± SE. * *p* < 0.05, ** *p* < 0.01, *** *p* < 0.001. ns indicates no significant difference between groups.

**Figure 2 ijms-23-05292-f002:**
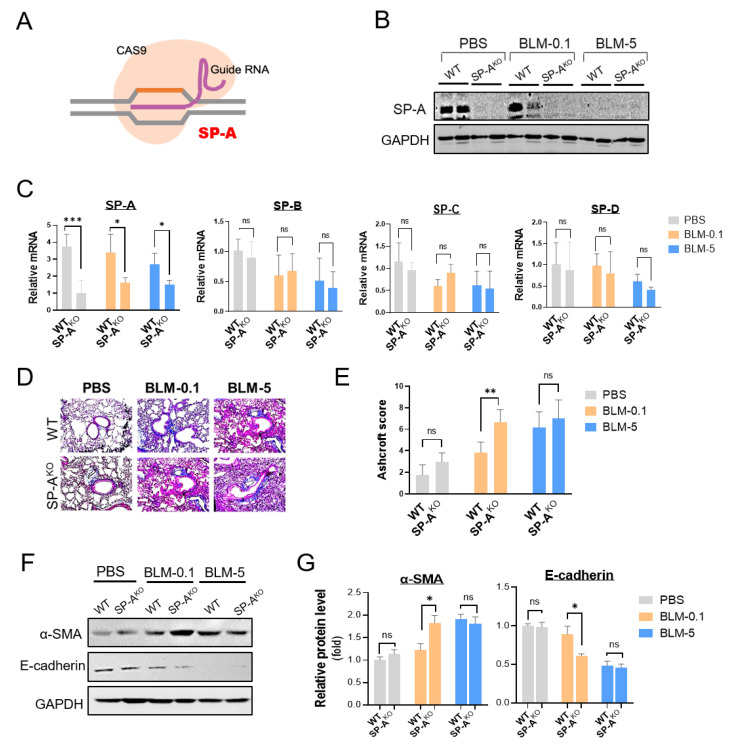
Lung surfactant SP-A deficiency promotes the development of pulmonary fibrosis. (**A**) Schematic showing CRISPR/Cas9-mediated genetic deletion of SP-A. SP-A deletion was performed by intratracheal delivery of lentiviral sgRNA targeting SP-A in CRISPR/Cas9 Rosa26-Cas9 knock-in mice (SP-A^KO^ group). As a control group, WT mice were generated by delivery of control lentivirus sgRNA (WT group). (**B**) Western blot analysis of lung SP-A protein levels in SP-A-deficient mice induced with low-dose BLM (BLM-0.1: 0.1 mg/kg BLM) or high-dose BLM (BLM-5: 5 mg/kg BLM). (**C**) Quantitative real-time qPCR analysis of SP-A, SP-B, SP-C, and SP-D in the lungs of SP-A-deficient BLM-induced mice. The relative level of surfactant protein was normalized to β-actin and calculated as fold change compared to those in PBS-treated WT group. (**D**,**E**) Representative images and semiquantitative analysis of Ashcroft scoring of Masson’s stained lung tissues from SP-A-deficient BLM-challenged mice. (**F**,**G**) Western blot analysis of E-cadherin and α-SMA in the lung tissues of mice. Densitometric analysis was represented. The relative level of E-cadherin or α-SMA was normalized to GAPDH and calculated as fold change compared to those in PBS-treated WT group. The data are presented as the mean ± SE. *n* = 3–6 per group. * *p* < 0.05, ** *p* < 0.01, *** *p* < 0.001. ns indicates no significant difference between groups. WT group: control lentivirus-injected mice; SP-A^KO^ group: mice induced with lentivirus containing sgRNA against SP-A.

**Figure 3 ijms-23-05292-f003:**
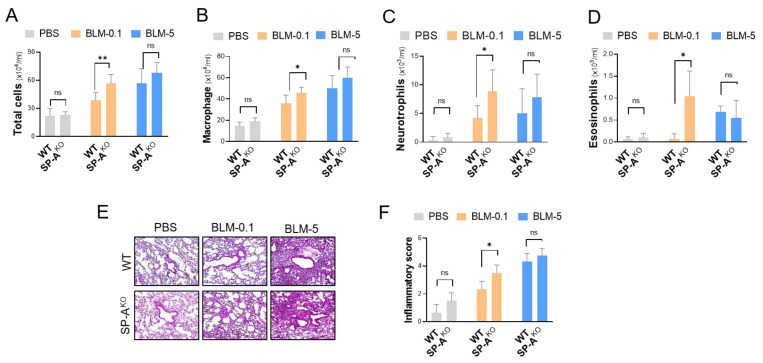
SP-A deficiency increases inflammatory cell accumulation in mice induced with a low dose of BLM. (**A**–**D**) Inflammatory cells in the BAL fluid of SP-A knockout mice following BLM treatment were counted to determine the total cell number (**A**), as well as the numbers of macrophages (**B**), neutrophils (**C**), and eosinophils (**D**). (**E**,**F**) Representative images of hematoxylin and eosin (H&E)-stained lung tissues and semiquantitative analysis of inflammation indices. The data are presented as the mean ± SE. * *p* < 0.05, ** *p* < 0.01. ns indicates no significant difference between groups. WT group: control lentivirus-injected mice; SP-A^KO^ group: lentivirus targeting SP-A-injected mice; BLM-0.1: 0.1 mg/kg BLM-induced mice; BLM-5: 5 mg/kg BLM-induced mice. *n* = 3–9 per group. WT group: control lentivirus-injected mice; SP-A^KO^ group: mice induced with lentivirus containing sgRNA against SP-A.

**Figure 4 ijms-23-05292-f004:**
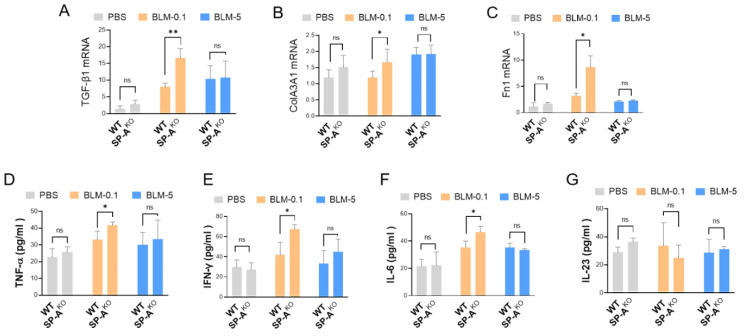
SP-A loss enhances the upregulation of fibrosis-associated genes and proinflammatory cytokines in the lungs of mice induced with a low dose of BLM. (**A**–**C**) The mRNA expression levels of TGF-β1 (**A**), Col3A1 (**B**), and Fn1 (**C**) in lung tissues of SP-A-deficient mice challenged with BLM (5 mg/kg BLM or 1 mg/kg BLM) were analyzed by quantitative real-time PCR. The relative level of these proteins was normalized to that of β-actin and calculated as fold change compared to those in PBS-treated WT group. (**D**–**G**) ELISA analysis of proinflammatory cytokines, such as TNF-α (**D**), IFN-γ (**E**), IL-6 (**F**), and IL-23 (**G**), in the lung tissues of SP-A-knockout mice following BLM administration. The data are presented as the mean ± SE. * *p* < 0.05, ** *p* < 0.01. ns indicates no significant difference between groups. WT group: control lentivirus-injected mice; SP-A^KO^ group: lentivirus targeting SP-A-injected mice; BLM-0.1: 0.1 mg/kg BLM-induced mice; BLM-5: 5 mg/kg BLM-induced mice. *n* = 3–9 per group.

**Figure 5 ijms-23-05292-f005:**
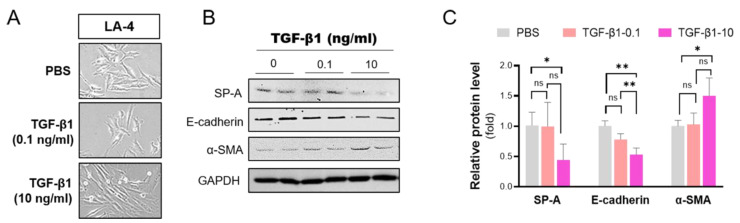
TGF-β1 downregulates SP-A levels in lung alveolar epithelial LA-4 cells. LA-4 cells were induced with various concentrations (0, 0.1, 10 ng/mL) of TGF-β1 for 24 h. (**A**) Morphological changes were observed under light microscopy (shown at 400ϗ magnification). (**B**,**C**) The protein levels of SP-A, E-cadherin, and α-SMA were evaluated by Western blotting (**B**) and subsequently analyzed by densitometry using ImageJ (**C**). The relative level of these proteins was normalized to GAPDH and calculated as fold change compared to those in PBS-treated WT group. The data are presented as the mean ± SE. * *p* < 0.05, ** *p* < 0.01. ns indicates no significant difference between groups. *n* = 3 per group.

**Figure 6 ijms-23-05292-f006:**
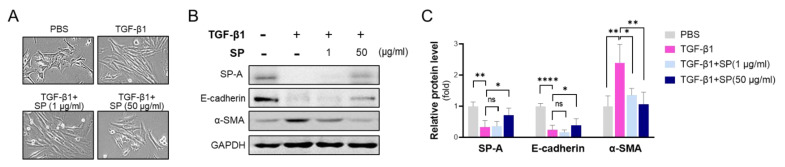
Exogenous treatment with surfactant protein suppresses TGF-β1-induced EMT in alveolar epithelial LA-4 cells. LA-4 cells were induced with TGF-β1 (20 ng/mL) for 24 h in the presence of isolated SP protein (1, 50 μg/mL). (**A**) Morphological changes were observed under light microscopy (shown at 400× magnification). (**B**,**C**) The protein levels of E-cadherin and α-SMA were evaluated by Western blotting (**B**) and subsequently analyzed by densitometry using ImageJ (**C**). The relative level of SP-A, E-cadherin, and α-SMA was normalized to GAPDH and calculated as fold change compared to those in PBS-treated WT group. The data are presented as the mean ± SE. * *p* < 0.05, ** *p* < 0.01, **** *p* < 0.0001. ns indicates no significant difference between groups. *n* = 4 per group.

**Table 1 ijms-23-05292-t001:** Oligonucleotides used in real-time PCR.

Gene	Direction	Sequence (5′-> 3′)
SP-A	Forward	gcagccaccctgagtttaga
	Reverse	ggcgaaaagagagcagttgg
SP-B	Forward	tagggcctcatccaaggtca
	Reverse	gatccagcatacactcggca
SP-C	Forward	atgaagctggcacatggact
	Reverse	ttttcccacgacgcctaaca
SP-D	Forward	tgctgccatacagcaactca
	Reverse	gctctcctgtggggtaagtg
TGF-β1	Forward	agcggactactatgctaaagaggtcaccc
	Reverse	ccaaggtaacgccaggaattgttgctata
Fn1	Forward	acagagctcaacctccctga
	Reverse	tgtgctctcctggttctcct
β-actin	Forward	gatctggcaccacaccttct
	Reverse	ggggtgttgaaggtctcaaa

## Data Availability

The raw data of the analyses presented in the figures in this study are available upon request from the corresponding author.

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
