# Peer review of "Loss of SP-A in the Lung Exacerbates Pulmonary Fibrosis"

_ijms, 2022, doi:10.3390/ijms23105292_

Round 1

Reviewer 1 Report

The authors evaluate the role of SP-A in experimental lung fibrosis. However, SP-A has already been proposed as biomarker of IPF, hence, in my opinion, this manuscript does not seem original.

In addition, too much mistakes appears throughtout the manuscript, making the reader loses interest. It is not understood how some figures does not match with the figure legend, for example Figure 2c is described as qPCR analysis, 2d and 2e as images, 2g is in the legend but no image is shown, 3f is explained as image of hematoxylin and eosin, Figure 5 is not placed correctly, etc.

Furthermore, section 2.1 and 2.2. of the results have the same title.

Lines 177-181 are explained as gene expression studies of proinflammatory cytokines refering the figure 4D-G. However, these images represent the protein levels of that cytokines assessed by ELISA, which could be different from mRNA expression of the genes that codes for that cytokines. This is a worrying mistake.

Section 4.9 in Material and Methods show that, besides SPA, other genes such as SPB, SPC and D are analyzed. However, no results are included in this respect.

Figure 1A-D should be considered as supplementary material given that seems to be more a confirmation of the BLM mouse model.

SMA should be explained in the introduction to understand its meaning in results. Results from that molecule should be included also in figure 1, given that the authors comment this result in line 99 in 2.1 section of the results.

References should not be included in results section, as well as to cite the figures in the discussion is not common. Furthermore, the last suggestions/interpretations included in results ´section would be more specific of the discussion.

All this are the main concerns that make the quality of presentation is very low, and, unfortunately, the manuscript would seem not to be reviewed to be submitted for publication.

Author Response

Response to Reviewer #1

We would like to thank the reviewer for careful and thorough reading of this manuscript and also for thoughtful comments and constructive suggestions, which help to improve the quality of this manuscript.

We agree with the reviewer’s suggestion and have made changes accordingly in Figure 1 and  Figure 2. We have accordingly rewritten sections of the manuscript so that it is now clearer and not make misleading or confusing. We hope that the reviewer finds these changes and this comply with the reviewer’s remarks.

Comment 1.      The authors evaluate the role of SP-A in experimental lung fibrosis. However, SP-A has already been proposed as biomarker of IPF, hence, in my opinion, this manuscript does not seem original.

Response 1.        We totally agree with the reviewer. SP-A has been proposed as a biomarker for IPF. Our main attempt was to identify the impact of lung SP-A, specifically in the development and progression of lung fibrosis.

Comment 2.       In addition, too much mistakes appears throughtout the manuscript, making the reader loses interest. It is not understood how some figures does not match with the figure legend, for example Figure 2c is described as qPCR analysis, 2d and 2e as images, 2g is in the legend but no image is shown, 3f is explained as image of hematoxylin and eosin, Figure 5 is not placed correctly, etc.

Furthermore, section 2.1 and 2.2. of the results have the same title.

Response 2.        We thank the reviewer for bringing up this point. We deeply apology for the error and corrected it. We have corrected the figure legend and rewritten result sections responding to reviewer comments in Figure 2 and Figure 3. Furthermore, we made changes the sentence in section 2.2 by replacing “SP-A-deficient mice displayed enhanced BLM-induced lung fibrosis in mice”.

Comment 3.       Lines 177-181 are explained as gene expression studies of proinflammatory cytokines refering the figure 4D-G. However, these images represent the protein levels of that cytokines assessed by ELISA, which could be different from mRNA expression of the genes that codes for that cytokines. This is a worrying mistake.

Response 3.        We agree that this was unclear and clarified the sentence with replacing “SP-A levels” by “SP-A protein levels” which is more accurate (Line 162).   

Comment 4.       Section 4.9 in Material and Methods show that, besides SPA, other genes such as SPB, SPC and D are analyzed. However, no results are included in this respect.

Response 4.        Thank you so much for pointing out. On initial submission, we put these data as supplementary figure. Here, we have added these data in Figure 2C and have accordingly expanded Figure 2.

Comment 5.       Figure 1A-D should be considered as supplementary material given that seems to be more a confirmation of the BLM mouse model.

Response 5.        We agree about this point. We thank you for this suggestion. We deleted these data in Figure 1A-D and presented them in a new Supplementary Figure S1. We agree about the reviewer’s comments. So we accordingly deleted most of parts about this Result section for the clarity on this study.

Comment 6.       SMA should be explained in the introduction to understand its meaning in results. Results from that molecule should be included also in figure 1, given that the authors comment this result in line 99 in 2.1 section of the results.

Response 6.        We apologize for this mistake. We thank you for thoughtful review. We have now added data from α-SMA western blotting in Result 2.1 section. Responding to reviewer’s comments, we also made changes as follows: “Therefore, we investigated the association of SP-A levels on BLM-induced lung fibro-sis with measuring protein level of fibrotic markers, such as α-smooth muscle actin (α-SMA). As expected, 5 mg/kg BLM-injected mice exhibited increased α-SMA. These data suggest and a negative correlation between α-SMA and SP-A on BLM-induced mice. Furthermore, we found loss of epithelial markers, such as E-cadherin, on high-dose BLM-induced mice, suggesting a positive correlation between E-cadherin and SP-A.” (Line 70-76)

Comment 7.       All this are the main concerns that make the quality of presentation is very low, and, unfortunately, the manuscript would seem not to be reviewed to be submitted for publication.

Response 7.        We agreed with this remark and apologized for the lack of clarity. We have now increased the resolution of the figures for publication.

Reviewer 2 Report

The authors analysed the beneficial effects of SP-A against lung fibrosis in vivo and in vitro.The manuscript was well written and the study design was correctly reported. Here the minor comments:

-Please state the limitation of the study

-improve the conclusion

Author Response

Response to Reviewer #2

We would like to thank the reviewer for careful and thorough reading of this manuscript and also for positive and thoughtful comments. We acknowledge for the reviewer’s comments and have added the limitation of our study in the discussion. We also have rewritten conclusion part entirely.

Comment 1        -Please state the limitation of the study

Response 1         We thank you for this remark and made limitation of the study in Discussion section as follows: “There were several limitations in the current study. First, we investigated the impact of SP-A on BLM-induced lung fibrosis without further evaluation of the underlying mechanism. Second, we performed biochemical experiments on only 14 days post-BLM. Thus, the impact of SP-A on lung fibrosis can be different based on the developing phases in lung fibrosis. Third, we treated SP protein which was purified from mouse BAL fluid on lung epithelial cells against TGF-β1. The therapeutic effect of synthetic SP-A can be different from that of isolated SP protein.” (Line 224-230)

Comment 2        -improve the conclusion

Response 2         We thank you for this suggestion. We have now made changes in the conclusion of Discussion section as follows: “In conclusion, this study demonstrates that SP-A loss exacerbated lung fibrosis induced by BLM in mice. The loss of SP-A protein expression was characterized by fibrotic change in the lung on BLM-treated mice. In addition, we also demonstrate that exogenous SP treatment effectively reversed TGF-β1-induced EMT in vitro. These data have significant implications for future efforts in developing a novel therapeutic strategy for treating IPF by targeting SP-A functions. Future experiments are needed to understand the mechanism underlying the impact of SP-A against IPF.” (Line 231-Line237)

Round 2

Reviewer 1 Report

The concern regarding the confusion between gene expression studies and protein studies remains to be solved (new lines 122-124). The authors performed ELISA analysis not gene expression for TNF-a, IFN-g, IL-6 and IL-23. The right sentences would be:

`Next, we examined whether SP-A deficiency caused alterations in the protein levels of proinflammatory cytokines in the lung tissues of mice. Indeed, we observed a notable increase in TNF-a, IFN-g,... ´

Author Response

Response to Reviewer #1

We sincerely appreciate the reviewer for thorough reading of this manuscript. We have revised the manuscript accordingly.

Comment 1.      The concern regarding the confusion between gene expression studies and protein studies remains to be solved (new lines 122-124). The authors performed ELISA analysis not gene expression for TNF-a, IFN-g, IL-6 and IL-23. The right sentences would be:

`Next, we examined whether SP-A deficiency caused alterations in the protein levels of proinflammatory cytokines in the lung tissues of mice. Indeed, we observed a notable increase in TNF-a, IFN-g,…

Response 1.        We thank the reviewer for bringing up this point. We deeply apology for the error and corrected it. We made changes this sentence by replacing “Next, we examined whether SP-A deficiency caused alterations in the protein levels of proinflammatory cytokines in the lung tissues of mice”.

Round 3

Reviewer 1 Report

The sentence 124-125 (´Indeed, we observed a notable increase in the gene expression of proinflammatory cytokines)  is still wrong.

The increase that you see is not in the gene expression. You are performing ELISA analysis to measure protein levels, not gene expression.

Author Response

We apologize for a mistake. We have revised the manuscript accordingly.

Comment 1.      The sentence 124-125 (´Indeed, we observed a notable increase in the gene expression of proinflammatory cytokines) is still wrong.

The increase that you see is not in the gene expression. You are performing ELISA analysis to measure protein levels, not gene expression.

Response 1.        We made changes this sentence by replacing “Indeed, we observed a notable increase in proinflammatory cytokines, including TNF-α, IFN-β, and IL-6 but not IL-23 in SP-A-deficient mice challenged with a low dose of BLM (Figure 4D-G).”

Round 4

Reviewer 1 Report

OK